statistics

interpretation of parameters, logistic model, missing values, model sensitivity

**Author for correspondence:**
H. S. Battey
e-mail: h.battey@imperial.ac.uk

# On the linear in probability model for binary data

## H. S. Battey[1], D. R. Cox[2] and M. V. Jackson[3]

[1]Department of Mathematics, Imperial College London, London, UK
[2]Nuffield College, Oxford, UK
[3]Department of Sociology, Stanford University, Stanford, CA, USA

HSB, 0000-0001-9387-4628

The analysis of binary response data commonly uses models linear in the logistic transform of probabilities. This paper considers some of the advantages and disadvantages of simple least-squares estimates based on a linear representation of the probabilities themselves, this in particular sometimes allowing a more direct empirical interpretation of underlying parameters. A sociological study is used in illustration.

## 1. Introduction

The interpretation of data in the form of binary outcomes arises in many areas of science from the primary physical and biological sciences and their application through to more directly applied areas and the social sciences.

Two distinct themes in the analysis of binary data go back at least to the beginning of the twentieth century with the contrast between Karl Pearson who, in his biserial correlation coefficient, treated a pair of possibly related binary variables as derived from an unobserved bivariate normally distributed variable, and Yule who worked directly with observed proportions of outcomes. When the hypothesized latent variables have a tangible interpretation, as in quantal bioassays, the former approach is preferable, but in the present paper we consider only situations in which observed proportions of outcomes are represented directly and relations concerning them interpreted.

Suppose that for $n$ independent individuals, we observe a realization of a binary outcome variable $Y_i$ ($1 \leq i \leq n$) taking values 1 or $-1$, and that for individual $i$ there is a $p \times 1$ vector $x_i$ of explanatory variables. A widely used representation is the linear logistic form in which $\log\{\mathrm{pr}(Y_i = 1)/\mathrm{pr}(Y_i = -1)\}$ is assumed to depend linearly on $x_i$. This leads to a simple interpretation of regression coefficients as ratios of effects when the binary responses are concentrated at one of the two levels but otherwise the interpretation is less direct. For a discussion from a sociological perspective of the difficulties of interpreting logistic coefficients, see [1] and, for a wide-ranging review, see [2].

The linear in probability model to be considered in the present paper specifies the probabilities as linear functions of the

explanatory variables, that is for $y = -1, 1$ and with $x_i$ typically including a constant term

$$\text{pr}(Y_i = y) = p_\beta(y) = \frac{1}{2}(1 + y\beta^T x_i), \tag{1.1}$$

so that $E(Y_i) = \beta^T x_i$. There are implicit restrictions on the parameter space, namely that for all data $x$, $|\beta^T x| \leq 1$.

If both the linear in probability and linear logistic models give adequate fit, the former has the advantage that the linear regression coefficients have a clearer operational interpretation in terms of numbers of individuals potentially influenced by a unit change of an explanatory variable. Emphasis sometimes lies on testing the significance of individual effects and comparison of their relative magnitudes. For this, the exponential family form of the linear logistic model [3,4] brings substantial simplification and other advantages. Furthermore, the logistic dependence has the potential to apply over a wide range of future conditions excluded by the positivity constraints on the linear form.

The discussion highlights a context in which maximum-likelihood estimation is very sensitive to aberrant observations, whereas ordinary least squares is insensitive yet typically achieves high efficiency.

A limiting case which sharply illustrates these distinctions concerns the comparison of data $(Y_1, Y_2)$ formed from counts of events from two Poisson processes of rates, say, $\rho_1$ and $\rho_1 \psi$ or $\rho_1$ and $\rho_1 + \theta$ for the multiplicative and additive representations, respectively. That is, $Y_2$ represents either a multiplication of the baseline rate by a constant or the addition of a separate signal. The former model falls within the exponential family of distributions and leads to an analysis based on a $2 \times 2$ contingency table. The second calls for a different analysis based on large-sample maximum-likelihood theory. For a further discussion concerning a similar model for Poisson variables, see [5].

# 2. Inferential aspects

## 2.1. Second-moment theory

We now consider properties of the linear in probability model based only on first and second moments. First, we define the least-squares estimate of $\beta$ by projecting the vector $Y = (Y_1, \ldots, Y_n)^T$ orthogonally onto the space spanned by the columns of $x$, thus giving

$$\hat{\beta}_{OLS} = (x^T x)^{-1} x^T Y.$$

In the present context, $x$ is a matrix whose $i$th row is $x_i^T$. The estimate is unbiased but does not have second-moment optimality unless $\beta = 0$ because the components of $Y$ in general do not have equal variance. Nor is the covariance matrix of the estimates given by the standard formulae unless $\beta$ is small. In fact

$$\text{var}(\hat{\beta}_{OLS}) = \Sigma_\beta = (x^T x)^{-1} - (x^T x)^{-1} x^T \Delta x (x^T x)^{-1}, \tag{2.1}$$

where $\Delta = \text{diag}(x_i^T \beta)^2$. One simple and often satisfactory estimate of the covariance matrix of $\hat{\beta}_{OLS}$ is to replace $\Delta$ by $\hat{\Delta}$ in which $\beta$ is replaced by $\hat{\beta}_{OLS}$.

A more elaborate second moment approach is to replace $\hat{\beta}_{OLS}$ by a weighted least-squares estimate $\hat{\beta}_{WLS}$ in which $\text{var}(Y_i)$ is estimated as $1 - (x_i^T \hat{\beta}_{OLS})^2$. Since $1 - (x_i^T \beta)^2$ is not bounded away from zero, weighted least squares is inappropriate as a general method.

The calculation of approximate confidence intervals and significance tests may be based on the asymptotic normality of $\hat{\beta}_{OLS}$.

## 2.2. Maximum-likelihood estimation

The log likelihood corresponding to (1.1) is

$$\ell(\beta) = \sum \log(1 + x_i^T \beta Y_i) \tag{2.2}$$

provided that for all $i$, $-1 < x_i^T \beta < 1$. We return to the relevance of this condition later. A stationary value of the log likelihood occurs where

$$\sum \frac{x_i Y_i}{1 + x_i^T \hat{\beta}_{ML} Y_i} = 0.$$

If $1/(1 + a)$ is expanded as $1 - a$ and higher terms neglected, that is the regression assumed small, the least-squares estimate $\hat{\beta}_{OLS}$ is recovered.

There is a strong argument for using ordinary least squares rather than maximum likelihood in this context despite sufficiency of $p_{\hat{\beta}_{ML}}$ under model (1.1). In the present context, the two estimators are virtually equivalent in terms of their efficiency, while maximum likelihood suffers extreme fragility, as explained below.

There is the following expansion of the second derivative of $\ell(\beta)$, valid for small $x_i^T \beta$,

$$\nabla_{\beta\beta}\ell(\beta) = -\sum \frac{x_i x_i^T Y_i^2}{(1 + x_i^T \beta Y_i)^2} = -\sum_i x_i x_i^T (1 - 2x_i^T \beta Y_i + 3(x_i^T \beta)^2) + O\{(x_i^T \beta)^3\}.$$

Here $\nabla_{\beta\beta}$ denotes the matrix of second partial derivatives with respect to $\beta$. On taking expectations, an approximation to the asymptotic variance of the maximum-likelihood estimator is obtained as $\{x^T(I + \Delta)x\}^{-1}$. For comparison to (2.1), it is more convenient to work with $\{x^T(I - \Delta)^{-1}x\}^{-1}$, which is a lower bound for $\{x^T(I + \Delta)x\}^{-1}$. Using the geometric series expansion $(I - \Delta)^{-1} = I + \Delta + \Delta^2 + \cdots = I + Y$, say, and the formula

$$(A + BC)^{-1} = A^{-1} - A^{-1}B(I + CA^{-1}B)^{-1}CA^{-1}, \tag{2.3}$$

we write, with $A = x^T x$, $B = I$ and $C = x^T Y x$ in (2.3) and $M = \{I + (x^T Y x)(x^T x)^{-1}\}^{-1}$,

$$\text{var}(\hat{\beta}_{ML}) = (x^T x)^{-1}\{I - M(x^T Y x)(x^T x)^{-1}\}. \tag{2.4}$$

Because $M \prec I$, where the notation $A \prec B$ means that $A - B$ is a negative definite matrix, the inflation in variance from using $\hat{\beta}_{OLS}$ rather than $\hat{\beta}_{ML}$ is

$$(x^T x)^{-1}\{(M - I)x^T \Delta x(x^T x)^{-1} + Mx^T(Y - \Delta)x(x^T x)^{-1}\} \prec (x^T x)^{-1}x^T(Y - \Delta)x(x^T x)^{-1}.$$

Write $\delta_i = \beta^T x_i$. From the geometric series, we deduce that

$$Y - \Delta = \text{diag}\left\{\frac{\delta_1^4}{(1 - \delta_1^2)}, \ldots, \frac{\delta_n^4}{(1 - \delta_n^2)}\right\}.$$

Thus $\text{var}(\hat{\beta}_{OLS}) - \text{var}(\hat{\beta}_{ML}) = O(n^{-1} \max\{\delta_i^4/(1 - \delta_i^2)\})$ showing that the loss in efficiency is typically very small.

On the other hand, from the perspective of formal likelihood theory even one individual out of range, in the sense that $|\beta^T x_i| > 1$, would refute the parameter value in question. That is, maximum likelihood is extremely sensitive in the present context to observations measured with error or drawn from a model even slightly different from that postulated. Ordinary least squares is by contrast relatively unaffected by such anomalies.

## 2.3. Interpretation of analysis

The interpretation of the regression coefficients in the linear in probability model is similar to that in a normal theory linear regression model. Let $x^*$ and $x^{**}$ be two different vectors of covariate information, differing by 1 unit in variable $j$ and otherwise the same. The number of positive outcomes is $S = \sum_i Z_i$ where $Z_i = (Y_i + 1)/2$. Therefore, the hypothetical change in $E(S)$ for a hypothetical replacement of $m$ individuals who differ by one unit in the $j$th component but are otherwise the same is

$$\sum_{i=1}^m \{E(Z_i \mid x^*) - E(Z_i \mid x^{**})\} = \frac{m\beta_j}{2}.$$

If there are binary covariates, it is natural to code them as $\{-1, 1\}$, in which case division of two is not needed because a unit change in the level corresponds to a numerical difference of two units.

If, upon fitting the linear in probability model, it is found that the number of least-squares fitted values $x_i^T \hat{\beta}_{OLS}$ outside $[-1, 1]$ is appreciably larger than could be attributed to chance under the linear in probability model, some doubt would be cast upon the plausibility of the model. The expected number out of range, assuming that the linear in probability model is valid for all observations, is $\lambda = \sum_i \text{pr}(|x_i^T \hat{\beta}_{OLS}| > 1) = \sum_i p_i$ where, by the asymptotic normality of $\hat{\beta}_{OLS} - \beta$,

$$p_i \simeq \Phi\left\{\frac{-1 + \beta^T x_i}{\sqrt{(x_i^T \Sigma_\beta x_i)}}\right\} + \Phi\left\{\frac{-1 - \beta^T x_i}{\sqrt{(x_i^T \Sigma_\beta x_i)}}\right\} \quad (n \to \infty).$$

Thus, a predicted number of out of range values is an estimate of $\lambda$, obtained by replacing $\beta$ and $\Sigma_\beta$ by estimates in the expression for each $p_i$. A crude lower bound on the variance of the sum, $R$, of out of range

values is $\lambda$, obtained by incorrectly assuming that $R$ is approximately Poisson distributed for large $n$. The variance of $R$ is larger than $\lambda$ due to dependence between the summands, induced by $\hat{\beta}_{\mathrm{OLS}}$. In particular,

$$\mathrm{var}(R) = \sum_i p_i(1 - p_i) + \sum_{i \neq j}\{\mathrm{pr}(|\hat{\beta}_{\mathrm{OLS}}^{\mathrm{T}}x_i| > 1, |\hat{\beta}_{\mathrm{OLS}}^{\mathrm{T}}x_j| > 1) - p_i p_j\}. \tag{2.5}$$

Write

$$Z_i = \frac{(\hat{\beta}_{\mathrm{OLS}} - \beta)^{\mathrm{T}}x_i}{\sqrt{(x_i^{\mathrm{T}}\Sigma_\beta x_i)}}, \quad z_i = \frac{1 - \beta^{\mathrm{T}}x_i}{\sqrt{(x_i^{\mathrm{T}}\Sigma_\beta x_i)}},$$

so that $Z_i$ and $Z_j$ are bivariate normally distributed of zero means, unit variances and correlation coefficient

$$\rho_{ij} = \frac{x_i^{\mathrm{T}}\Sigma_\beta x_j}{\sqrt{(x_i^{\mathrm{T}}\Sigma_\beta x_i)}\sqrt{(x_j^{\mathrm{T}}\Sigma_\beta x_j)}}.$$

Then $\mathrm{pr}(|\hat{\beta}_{\mathrm{OLS}}^{\mathrm{T}}x_i| > 1, |\hat{\beta}_{\mathrm{OLS}}^{\mathrm{T}}x_j| > 1)$ is the sum of the quadrant probabilities,

$$\mathrm{pr}(Z_i > z_i, Z_j > z_j) = \Phi(-z_i)\int_{z_i}^\infty \Phi\left\{\frac{\rho_{ij}s - z_j}{\sqrt{(1 - \rho_{ij}^2)}}\right\}\Phi(s)\,\mathrm{d}s,$$

$$\mathrm{pr}(Z_i < -z_i, Z_j < -z_j) = \Phi(-z_i)\int_{-\infty}^{-z_i} \Phi\left\{\frac{-z_j + \rho_{ij}s}{\sqrt{(1 - \rho_{ij}^2)}}\right\}\Phi(s)\,\mathrm{d}s,$$

$$\mathrm{pr}(Z_i > z_i, Z_j < -z_j) = \Phi(-z_i)\int_{z_i}^\infty \Phi\left\{\frac{-z_j - \rho_{ij}s}{\sqrt{(1 - \rho_{ij}^2)}}\right\}\Phi(s)\,\mathrm{d}s$$

and

$$\mathrm{pr}(Z_i < -z_i, Z_j > z_j) = \Phi(-z_j)\int_{z_j}^\infty \Phi\left\{\frac{-z_i - \rho_{ij}s}{\sqrt{(1 - \rho_{ij}^2)}}\right\}\Phi(s)\,\mathrm{d}s.$$

While there is no closed-form expression for these, close approximations are obtained by replacing the conditional expectations of the functions of interest by the corresponding functions of the conditional expectations, with approximation error established by Taylor series expansion. Depending on the signs of $z_i$, $z_j$ and $\rho_{ij}$, the approximation so obtained might be improved by interchanging the roles of $z_i$ and $z_j$ on the right-hand side of the above display. For a further discussion, see [6].

# 3. Socio-economic inequalities in educational attainment

We use US data from the National Longitudinal Study of Youth (1979), a nationally representative longitudinal study of people aged 14–22. Our binary outcome, coded as $\{-1, 1\}$, specifies whether the individual enrolled in a 4-year-degree-granting institution for at least 1 year. There are five potential explanatory variables. Ability is measured as the respondent's score on the Armed Forces Qualifying Test, administered to all respondents in the 1981 wave of the survey. Family income in childhood is measured as the log of total net family income in 1979. All respondents identified themselves as male or female but race was measured via interviewer observation, and we here limit our sample to those respondents who were classified as black or non-black and non-Hispanic. Finally, we include an indicator of whether respondents were living with at least one parent at the time of the first survey.

As is common with extensive observational data, some observations on explanatory variables are missing, as shown in table 1. Because we are concerned with the dependence of outcome on explanatory variables, individuals with missing outcome are treated as uninformative about that dependence. A sensitivity analysis examined how the regression coefficients of interest changed when rather extreme assignments were made to the three explanatory variables with missing values, treating binary variables as all at one or other extreme and continuous variables as at their upper and lower quartile. The levels used were 68.33 and 17.28 for the Armed Forces Qualifying Test score and 10.00 and 8.79 for the logarithm of family income when the individual was in childhood. Estimates from the eight patterns of missingness are in table 2. While there is some dependence on the missing values, that dependence is very minor and without qualitative impact on the conclusions of the analysis. If a larger number of explanatory variables have missing values the sensitivity analysis should be based on a suitable fraction of the two-level factorial system of potential missing values, allowing estimation of main effects from missingness [7, §12.2].

**Table 1.** Summary of data.

| covariate | description | sample range | per cent missing |
|---|---|---|---|
| $x_1$ | gender | $\{1 = \text{male}, -1 = \text{female}\}$ | 0 |
| $x_2$ | AFQT score | percentage $(0-100)$ | 4.3 |
| $x_3$ | log income | continuous $(3.00-11.23)$ | 51.2 |
| $x_4$ | race | $\{1 = \text{black}, -1 = \text{non-black/non-Hispanic}\}$ | 0 |
| $x_5$ | lives with parent | $\{1 = \text{yes}, -1 = \text{no}\}$ | 5.1 |

The sensitivity analysis used here may be contrasted with procedures of multiple imputation based on the untestable assumption that observations are missing at random.

An informal preliminary analysis involved tests for interactions and inspection of interaction plots. None was strongly suggested. Table 2 reports least squares estimates of regression coefficients and their estimated standard errors from a model with main effects for the five explanatory variables.

The suggestion is that hypothetically increasing the number of males and correspondingly reducing the number of females in the population by $m$ units, say, would correspond to a 6–7% of $m$ decrease in the expected number of individuals receiving higher education, all other things equal. The coefficient of the race variable is similarly interpreted, the suggestion being that in a hypothetical population, demographically equivalent to the one under study except for having $m$ more black children than white children, the expected number of individuals experiencing the positive outcome would be 22–23% higher.

It is suggested, all other things being equal, that a 1% increase in family income, i.e. an increase of 0.01 in log family income, would correspond to a 0.02–0.03% increase in the expected number of positive outcomes and that a 1% increase in ability, to the extent that it can be measured by the Armed Forces Qualifying Test score, would correspond to a 1% increase. An absolute change at the bottom of the income scale has a relatively greater effect than the same absolute change at the top. Finally, accounting for other factors, individuals living with someone other than one of their parents are perhaps slightly more likely to experience the positive outcome, although the evidence for this is rather weak.

In the above interpretation of the estimated coefficients on the continuous variables, division by 2 is needed, as described in §2.3. Division by 2 is not needed for the three binary explanatory variables because they are coded as $\{-1, 1\}$.

The last two columns of table 2 show the actual and predicted number of least squares fitted values $x_i^{\mathrm{T}} \hat{\beta}_{\mathrm{OLS}}$ that are outside $[-1, 1]$. The individuals whose fitted values are out of range are almost all at the two edges of the sample space for the Armed Forces Qualifying Test score.

While the numerical values of the coefficient estimates from a linear logistic model are not comparable to those from a linear in probability model, the ratios of these coefficients are remarkably similar. The code for verifying this statement and the analysis of §3 is available as outlined in the data accessibility statement.

# 4. Discussion

As with other statistical methods care is needed especially when relatively complex data are involved. In the present context, a reasonable approach for general use is to base the analysis on $\hat{\beta}_{\mathrm{OLS}}$ with the improved estimate of its covariance matrix, given by (2.1). Examination of model adequacy should include a check of the number of fitted values outside $[-1, 1]$. Do such values form a rationally identifiable subgroup to be analysed separately? Does their omission or exclusion materially affect the conclusions? Does the number of anomalous observations suggest major change to the whole analysis? A large number of anomalous observations may suggest that a model linear on the logit scale would be more appropriate.

From the perspective of formal likelihood theory, even one individual out of range would refute the parameter value in question in the linear in probability model. Thus, the paper illustrates an empirical context in which the formal optimality of maximum-likelihood estimates is achieved only at the cost of extreme fragility. A formally slightly less efficient method is much to be preferred.

**Table 2.** Sensitivity analysis of least squares estimates and their estimated standard errors from replacing all missing values of $x_j$ by high and low levels. The estimated standard errors are obtained by replacing $\Delta$ by $\hat{\Delta}$ in equation (2.1). The sample size is 9043.

| $x_2$ | $x_3$ | $x_5$ | least squares estimates of regression coefficients (estimated standard errors) | | | | | | number out of range | predicted number out of range |
| --- | --- | --- | --- | --- | --- | --- | --- | --- | --- | --- |
| | | | $\hat{\beta}_0$ | $\hat{\beta}_1$ | $\hat{\beta}_2$ | $\hat{\beta}_3$ | $\hat{\beta}_4$ | $\hat{\beta}_5$ | | |
| L | H | H | −1.51 (0.13) | −0.061 (0.0092) | 0.0201 (0.00031) | 0.064 (0.011) | 0.224 (0.011) | −0.034 (0.011) | 394 | 396 |
| L | H | L | −1.51 (0.13) | −0.062 (0.0092) | 0.0202 (0.00031) | 0.063 (0.014) | 0.223 (0.011) | −0.021 (0.010) | 383 | 388 |
| L | L | H | −1.32 (0.12) | −0.060 (0.0092) | 0.0202 (0.00031) | 0.048 (0.014) | 0.222 (0.011) | −0.038 (0.011) | 384 | 391 |
| L | L | L | −1.31 (0.12) | −0.061 (0.0092) | 0.0203 (0.00031) | 0.046 (0.014) | 0.221 (0.011) | −0.025 (0.011) | 377 | 384 |
| H | H | H | −1.57 (0.13) | −0.065 (0.0093) | 0.0198 (0.00033) | 0.068 (0.014) | 0.225 (0.011) | −0.028 (0.011) | 444 | 441 |
| H | H | L | −1.57 (0.13) | −0.067 (0.0094) | 0.0198 (0.00032) | 0.066 (0.014) | 0.223 (0.011) | −0.011 (0.010) | 434 | 436 |
| H | L | H | −1.45 (0.13) | −0.065 (0.0094) | 0.0198 (0.00033) | 0.059 (0.014) | 0.224 (0.011) | −0.034 (0.012) | 451 | 450 |
| H | L | L | −1.44 (0.13) | −0.066 (0.0094) | 0.0199 (0.00033) | 0.056 (0.014) | 0.222 (0.011) | −0.017 (0.011) | 453 | 443 |
| max absolute difference | | | 0.23 | 0.0061 | 0.00050 | 0.022 | 0.0040 | 0.026 | | |

Data accessibility. The data and code used in the example, together with further details about the data source, can be accessed from the Royal Society's repository: https://rs.figshare.com.

Authors' contributions. H.S.B., D.R.C. and M.V.J. designed the research, performed the research and wrote the paper.

Competing interests. The authors confirm that there are no competing interests.

Funding. The work was supported by the UK Engineering and Physical Sciences Research Council under grant no. EP/P002757/1 and by the Russell Sage Foundation and the Andrew W. Mellon Foundation Fellowship at the Center for Advanced Study in the Behavioral Sciences, Stanford University.

Acknowledgements. We thank the referees for constructive comments.

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
