## [Reviewer comments · Royal Society Open Science]

Review History

RSOS-190067.R0 (Original submission)

Review form: Reviewer 1

Is the manuscript scientifically sound in its present form?

Yes

Are the interpretations and conclusions justified by the results?

Yes

Is the language acceptable?

Yes

Is it clear how to access all supporting data?

No

Do you have any ethical concerns with this paper?

No

Have you any concerns about statistical analyses in this paper?

Yes

Recommendation?

Accept with minor revision (please list in comments)

Comments to the Author(s)

This paper might be seen as expanding Section 2.2 of Cox (1970, Binary Data, Chapman and Hall); some other parts of the same chapter also seem relevant. I find it a bit surprising that this book and the second edition with Snell (1989) are not cited, particularly as the authors of the present work seem to contradict advice therein about the linear in probabilities model: 'use of a model, the nature of whose limitations can be foreseen, is not wise, except for very restricted purposes' (Cox and Snell, 1989, page 14).

Apart from this, my comments are relatively minor:

page 1, line 59 (using the numbers at the left of the page): 'interpretation is more formal'. Meaning? And what are 'the data on Y_i '? I thought the y_i s were the data. This entire passage is obscure (including the bit about concentrated at the end of the scale---the scale is \pm , so this presumably means the data are mostly -1s or 1s?)

page 2, line 15-21: The second may also involve a small sample analysis, as in the more elaborate but similar model discussed in Davison and Sartori (2008, Statistical Science).

page 2, line 44-45: simplify to ... tests may be based on ...

page 3, line 33: bracket trouble

page 3, lines 35-39: What's the evidence that OLS is less sensitive to bad data? Usually it is regarded as the paradigm of a non-robust estimation procedure (though here this is mitigated by the binary nature of the responses). The sensitivity alluded to might be regarded as a strong argument in favour of the logistic model ...

page 4, lines 28-38: Surely there are closed forms for these expressions, since the Φ parts can be expressed as a conditional probability given the variable in the ϕ part, and the whole reduces to a Φ ?

page 5, line 1: 'explanatory observations'?

page 5, line 32: the final analysis results seem to be missing from Table 2. (Or, if not, to which line do they correspond?) Generally there seems to be some confusion in the text about the contents of this table (and its caption seems inadequate---what sort of estimates, for example?).

Table 2: could we see (and could you please briefly discuss) the results of a logistic regression analysis, for comparison? If a linear Taylor series expansion around the MLEs of the probabilities was used, would it lead to essentially the same conclusions? It would also be useful to be able to compare the SEs for the ML, LS and logistic models.

What actually is the empirical value of the efficiency loss shown at page 3, line 33, in this example?

page 7, line 40: Agresti (he is not aggressive)

Review form: Reviewer 2 (John MacInnes)

Is the manuscript scientifically sound in its present form?

Yes

Are the interpretations and conclusions justified by the results?

Yes

Is the language acceptable?

Yes

Is it clear how to access all supporting data?

Yes

Do you have any ethical concerns with this paper?

No

Have you any concerns about statistical analyses in this paper?

No

Recommendation?

Accept with minor revision (please list in comments)

Comments to the Author(s)

I leave it to others better qualified than me to comment on the formal statistical argument of this paper. However I do have two observations relevant to its implications and application.

Binary response data is ubiquitous in the social sciences. Students and researchers are taught that to fit a linear model is wrong because residuals cannot be normally distributed and estimates of the response variable can be out of range. I plead guilty to being one of many authors to argue this. Because of this, logistic regression has become a workhorse underpinning published research across the social sciences. However, criticism of the linear model has not always been accompanied by adequate appreciation of the difficulties of interpreting logistic coefficients because of the way in which they are confounded with residual variation. This complicates the comparison of coefficients across nested models of the same group of observations, or between the same model applied to different groups, since coefficients will be biased towards zero but by different amounts, since residual variation is unlikely to be constant across models, as the references [1] [2] cited by the authors point out. Although this problem has been discussed in the methodological literature for over twenty years, awareness of it among applied social scientists, journal editors and referees has lagged behind.

In this context, the suggestion that a linear in probability model may sometimes be preferred to the use of logistic transformation of the probabilities for binary response data has enormous significance. The upshot of this paper is that the advice routinely given to students and researchers in social statistics is at best incomplete, and in some circumstances simply wrong. The authors may feel that it is not their job to translate the conclusions of their paper into advice or guidance aimed at those unable to follow in full the presentation of their argument. It could also be argued that any such advice carries the danger that it might, by encouraging researchers to use a procedure they do not fully understand, support poor statistical work. However, I would encourage the authors to consider setting out some of their conclusions in a form accessible to a wider audience than it might currently reach. While they may feel that, for example, to compare the results of the empirical analysis presented here with that produced by logistic regression would be pedestrian, whereas I think it would help many readers to appreciate the way in which

the approach described here produces a set of coefficients that are less sensitive to patterns in missingness and with much smaller standard errors than those produced by logistic transformation. Doing so might help to spread awareness much more quickly beyond the methodologists to those doing and refereeing applied work.

I suspect readers with a social science background will also find the interpretation of relations concerning the observed proportion of outcomes surprising. Logistic regression coefficients are typically interpreted in terms of the relative odds of a positive outcome. In the model presented here, because it is the proportion of outcomes itself that is the dependent, the conclusion is presented in terms 'numbers of individuals potentially influenced by a unit change of an explanatory variable', that is, shifting from one category to another of the outcome variable. The unit change in explanatory categorical variables is treated as a change in the characteristics of a hypothetical population, such as a change in the proportion of sexes or ethnic groups. While this is formally clear, its substantive interpretation is less obvious. Researchers may often be less interested in comparing outcomes across differing hypothetical populations than in inferring the relative odds of a positive outcome for different groups within them. It is not clear to me, and it would be helpful if the authors discussed this more explicitly, whether such inferences would be legitimate with this model. On page 2 the authors appear to continue to favour a logistic approach here ('testing the significance of individual effects'). However, in the discussion of the interpretation of the explanatory variables, the example of family circumstances is treated in terms of the relative probability of experiencing a positive outcome. This confusion is my responsibility rather than that of the authors, but I suspect I would not be the only reader who would benefit from a slightly longer exposition of this point.

One very minor point. The authors adequately describe the source of the data used in the illustration and its subsequent treatment. However my understanding is that good reproducibility practice would be to include an r script or other code of the model run and the sensitivity analysis made.

John MacInnes
University of Edinburgh

Review form: Reviewer 3

Is the manuscript scientifically sound in its present form?

Yes

Are the interpretations and conclusions justified by the results?

Yes

Is the language acceptable?

Yes

Is it clear how to access all supporting data?

Yes

Do you have any ethical concerns with this paper?

No

Have you any concerns about statistical analyses in this paper?

No

Recommendation?

Accept with minor revision (please list in comments)

Comments to the Author(s)

The authors investigate advantages and disadvantages of a linear in probability model for the analysis of binary data.

Discussions about advantages and disadvantages between the commonly used linear model in the logistic transform of probabilities and a simpler linear in probability model are recurrent in the literature of different fields. For instance, in the past decade there has been a heated debate in the economic literature about this question and more broadly about the need to go beyond simple linear models.

This is a refreshing paper and a useful guideline for statisticians and data analysts. A positive aspect of this paper is a clear discussion of the properties of OLS and MLE for the linear in probability model and the interpretation of the corresponding coefficients, as well as the introduction of a simple tool for model checking.

I have just a question about the sensitivity of MLE versus OLS; see the comment in the middle of p. 3. The sensitivity of the MLE is already apparent from the score function in the estimating equation below formula (3) at p. 2. This function becomes very large for x 's such that $x^T\beta$ is close to the boundary 1, making the influence function of the MLE unbounded. In the paper it is stated that "OLS is by contrast relatively unaffected ..." (p. 3), but I do not see the reason. In fact its score function will always be affected by large values of some covariates.

Decision letter (RSOS-190067.R0)

21-Mar-2019

Dear Dr Battey

On behalf of the Editors, I am pleased to inform you that your Manuscript RSOS-190067 entitled "On the linear in probability model for binary data" has been accepted for publication in Royal Society Open Science subject to minor revision in accordance with the referee suggestions. Please find the referees' comments at the end of this email.

The reviewers and handling editors have recommended publication, but also suggest some minor revisions to your manuscript. Therefore, I invite you to respond to the comments and revise your manuscript.

- Ethics statement

- Data accessibility

It is a condition of publication that all supporting data are made available either as supplementary information or preferably in a suitable permanent repository. The data accessibility section should state where the article's supporting data can be accessed. This section

should also include details, where possible of where to access other relevant research materials such as statistical tools, protocols, software etc can be accessed. If the data has been deposited in an external repository this section should list the database, accession number and link to the DOI for all data from the article that has been made publicly available. Data sets that have been deposited in an external repository and have a DOI should also be appropriately cited in the manuscript and included in the reference list.

If you wish to submit your supporting data or code to Dryad (<http://datadryad.org/>), or modify your current submission to dryad, please use the following link:
<http://datadryad.org/submit?journalID=RSOS&manu=RSOS-190067>

- **Competing interests**

- **Authors' contributions**

- **Acknowledgements**

- **Funding statement**

Because the schedule for publication is very tight, it is a condition of publication that you submit the revised version of your manuscript before 30-Mar-2019. Please note that the revision deadline will expire at 00.00am on this date. If you do not think you will be able to meet this date please let me know immediately.

To revise your manuscript, log into <https://mc.manuscriptcentral.com/rsos> and enter your Author Centre, where you will find your manuscript title listed under "Manuscripts with Decisions". Under "Actions," click on "Create a Revision." You will be unable to make your

revisions on the originally submitted version of the manuscript. Instead, revise your manuscript and upload a new version through your Author Centre.

Once again, thank you for submitting your manuscript to Royal Society Open Science and I look

forward to receiving your revision. If you have any questions at all, please do not hesitate to get in touch.

on behalf of Professor Ruth King (Associate Editor) and Professor Mark Chaplain (Subject Editor)
 openscience@royalsociety.org

Associate Editor Comments to Author (Professor Ruth King):

The paper has been reviewed by myself and three reviewers. Overall, the paper has been very well written and presents an interesting discussion of the use of the linear model for binary data, as opposed to the commonly used logistic regression model. There are a few minor issues which would be useful to address, as identified by the reviewers, particularly in relation to the comparison of the linear model with the logistic model in the example provided - see reviewers comments for additional comments. Although the appendix provides a description of how to access the data is it possible to obtain permission to be able to provide the data with the paper? Finally, it would also be useful to provide an R script for reproducing the results of the paper given the data.

Reviewer comments to Author:

Reviewer: 1

Comments to the Author(s)

This paper might be seen as expanding Section 2.2 of Cox (1970, Binary Data, Chapman and Hall); some other parts of the same chapter also seem relevant. I find it a bit surprising that this book and the second edition with Snell (1989) are not cited, particularly as the authors of the present work seem to contradict advice therein about the linear in probabilities model: 'use of a model, the nature of whose limitations can be foreseen, is not wise, except for very restricted purposes' (Cox and Snell, 1989, page 14).

Apart from this, my comments are relatively minor:

page 1, line 59 (using the numbers at the left of the page): 'interpretation is more formal'. Meaning? And what are 'the data on Y_i '? I thought the y_i s were the data. This entire passage is obscure (including the bit about concentrated at the end of the scale---the scale is \pm , so this presumably means the data are mostly -1s or 1s?)

page 2, line15-21: The second may also involve a small sample analysis, as in the more elaborate but similar model discussed in Davison and Sartori (2008, Statistical Science).

page 2, line 44-45: simplify to ... tests may be based on ...

page 3, line 33: bracket trouble

page 3, lines 35-39: What's the evidence that OLS is less sensitive to bad data? Usually it is regarded as the paradigm of a non-robust estimation procedure (though here this is mitigated by the binary nature of the responses). The sensitivity alluded to might be regarded as a strong argument in favour of the logistic model ...

page 4, lines 28-38: Surely there are closed forms for these expressions, since the Φ parts can be expressed as a conditional probability given the variable in the ϕ part, and the whole reduces to a Φ ?

page 5, line 1: 'explanatory observations'?

page 5, line 32: the final analysis results seem to be missing from Table 2. (Or, if not, to which line do they correspond?) Generally there seems to be some confusion in the text about the contents of this table (and its caption seems inadequate---what sort of estimates, for example?).

Table 2: could we see (and could you please briefly discuss) the results of a logistic regression analysis, for comparison? If a linear Taylor series expansion around the MLEs of the probabilities was used, would it lead to essentially the same conclusions? It would also be useful to be able to compare the SEs for the ML, LS and logistic models.

What actually is the empirical value of the efficiency loss shown at page 3, line 33, in this example?

page 7, line 40: Agresti (he is not aggressive)

Reviewer: 2

Comments to the Author(s)

I leave it to others better qualified than me to comment on the formal statistical argument of this paper. However I do have two observations relevant to its implications and application.

Binary response data is ubiquitous in the social sciences. Students and researchers are taught that to fit a linear model is wrong because residuals cannot be normally distributed and estimates of the response variable can be out of range. I plead guilty to being one of many authors to argue this. Because of this, logistic regression has become a workhorse underpinning published research across the social sciences. However, criticism of the linear model has not always been accompanied by adequate appreciation of the difficulties of interpreting logistic coefficients because of the way in which they are confounded with residual variation. This complicates the comparison of coefficients across nested models of the same group of observations, or between the same model applied to different groups, since coefficients will be biased towards zero but by different amounts, since residual variation is unlikely to be constant across models, as the references [1] [2] cited by the authors point out. Although this problem has been discussed in the methodological literature for over twenty years, awareness of it among applied social scientists, journal editors and referees has lagged behind.

In this context, the suggestion that a linear in probability model may sometimes be preferred to the use of logistic transformation of the probabilities for binary response data has enormous significance. The upshot of this paper is that the advice routinely given to students and researchers in social statistics is at best incomplete, and in some circumstances simply wrong. The authors may feel that it is not their job to translate the conclusions of their paper into advice or guidance aimed at those unable to follow in full the presentation of their argument. It could also be argued that any such advice carries the danger that it might, by encouraging researchers to use a procedure they do not fully understand, support poor statistical work. However, I would encourage the authors to consider setting out some of their conclusions in a form accessible to a wider audience than it might currently reach. While they may feel that, for example, to compare

the results of the empirical analysis presented here with that produced by logistic regression would be pedestrian, whereas I think it would help many readers to appreciate the way in which the approach described here produces a set of coefficients that are less sensitive to patterns in missingness and with much smaller standard errors than those produced by logistic transformation. Doing so might help to spread awareness much more quickly beyond the methodologists to those doing and refereeing applied work.

I suspect readers with a social science background will also find the interpretation of relations concerning the observed proportion of outcomes surprising. Logistic regression coefficients are typically interpreted in terms of the relative odds of a positive outcome. In the model presented here, because it is the proportion of outcomes itself that is the dependent, the conclusion is presented in terms 'numbers of individuals potentially influenced by a unit change of an explanatory variable', that is, shifting from one category to another of the outcome variable. The unit change in explanatory categorical variables is treated as a change in the characteristics of a hypothetical population, such as a change in the proportion of sexes or ethnic groups. While this is formally clear, its substantive interpretation is less obvious. Researchers may often be less interested in comparing outcomes across differing hypothetical populations than in inferring the relative odds of a positive outcome for different groups within them. It is not clear to me, and it would be helpful if the authors discussed this more explicitly, whether such inferences would be legitimate with this model. On page 2 the authors appear to continue to favour a logistic approach here ('testing the significance of individual effects'). However, in the discussion of the interpretation of the explanatory variables, the example of family circumstances is treated in terms of the relative probability of experiencing a positive outcome. This confusion is my responsibility rather than that of the authors, but I suspect I would not be the only reader who would benefit from a slightly longer exposition of this point.

One very minor point. The authors adequately describe the source of the data used in the illustration and its subsequent treatment. However my understanding is that good reproducibility practice would be to include an r script or other code of the model run and the sensitivity analysis made.

John MacInnes
University of Edinburgh

Reviewer: 3

Comments to the Author(s)

The authors investigate advantages and disadvantages of a linear in probability model for the analysis of binary data.

Discussions about advantages and disadvantages between the commonly used linear model in the logistic transform of probabilities and a simpler linear in probability model are recurrent in the literature of different fields. For instance, in the past decade there has been a heated debate in the economic literature about this question and more broadly about the need to go beyond simple linear models.

This is a refreshing paper and a useful guideline for statisticians and data analysts. A positive aspect of this paper is a clear discussion of the properties of OLS and MLE for the linear in probability model and the interpretation of the corresponding coefficients, as well as the introduction of a simple tool for model checking.

I have just a question about the sensitivity of MLE versus OLS; see the comment in the middle of

p. 3. The sensitivity of the MLE is already apparent from the score function in the estimating equation below formula (3) at p. 2. This function becomes very large for x 's such that $x^T\beta$ is close to the boundary 1, making the influence function of the MLE unbounded. In the paper it is stated that "OLS is by contrast relatively unaffected ..." (p. 3), but I do not see the reason. In fact its score function will always be affected by large values of some covariates.

Author's Response to Decision Letter for (RSOS-190067.R0)

See Appendix A.

Decision letter (RSOS-190067.R1)

05-Apr-2019

Dear Dr Battey,

I am pleased to inform you that your manuscript entitled "On the linear in probability model for binary data" is now accepted for publication in Royal Society Open Science.

on behalf of Professor Ruth King (Associate Editor) and Mark Chaplain (Subject Editor)
openscience@royalsociety.org

Appendix A

Manuscript number RSOS-190067

We thank the associate editor and the three referees for their encouraging reports. The following changes reflect their suggestions.

- We have slightly enlarged the first paragraph of page 2 to clarify when the linear logistic model would be preferred to the linear in probability model and vice versa.
- Wording has been simplified in: the third paragraph of page 1; the final sentence of §2.1; the second paragraph of §3.
- Table 1’s caption and title has been clarified.
- A brief discussion of the linear logistic model has been added at the end of §3.

The additional points seem not to need major changes to the paper.

- The fragility of maximum likelihood in the context of the linear in probability model is due to the implicit restrictions on the parameter space. The empirical fitting by least squares ignores these restrictions. By accepting the possibility of out of range fitted values, the coefficient estimates are made less sensitive to aberrant observations. We feel that a more formal analysis in terms of influence functions is unnecessary.
- Closed forms for the integrals on page 4 do not seem available. Good analytic approximations are provided in reference [5].
- A brief discussion of the linear logistic model has been added at the end of §3. We decided not to include results of a logistic model as these would not be very enlightening. The analysis has been performed and is available in the code, which we have deposited in your data repository, along with the data used. Figure 1 summarizes the output of this analysis, which is as expected.

Figure: ratios of maximum likelihood estimates of the logistic regression coefficients plotted against the ratios of estimated coefficients for the same pairs of variables obtained by fitting the linear in probability model by least squares. The colour represents different pairs of coefficients, with duplicate colours corresponding to duplicate pairs from the eight patterns of missingness described by the rows of Table 1 in the paper.